# Hyaluronan-Arginine Interactions—An Ultrasound and ITC Study

**DOI:** 10.3390/polym12092069

**Published:** 2020-09-12

**Authors:** Adam Jugl, Miloslav Pekař

**Affiliations:** Faculty of Chemistry, Brno University of Technology, Purkyňova 464, 612 00 Brno, Czech Republic; xcjugl@fch.vut.cz

**Keywords:** hyaluronan, arginine, interaction, high-resolution ultrasound spectroscopy, isothermal titration calorimetry

## Abstract

High-resolution ultrasound spectroscopy and isothermal titration calorimetry were used to characterize interactions between hyaluronan and arginine oligomers. The molecular weight of arginine oligomer plays an important role in interactions with hyaluronan. Interactions were observable for arginine oligomers with eight monomer units and longer chains. The effect of the ionic strength and molecular weight of hyaluronan on interactions was tested. In an environment with increased ionic strength, the length of the arginine oligomer was crucial. Generally, sufficiently high ionic strength suppresses interactions between hyaluronan and arginine oligomers, which demonstrated interactions in water. From the point of view of the molecular weight of hyaluronan, the transition between the rod conformation and the random coil conformation appeared to be important.

## 1. Introduction

Hyaluronan is a naturally occurring polysaccharide, which was discovered by Meyer and Palmer in the vitreous humor of cattle eyes [1]. This biopolymer, with a relatively simple primary structure, is synthesized by organisms from bacteria to humans [2]. However, according to research results to date, despite its simple composition [3], hyaluronan plays a significant role in many biological processes of the human body, such as regulating water balance and osmotic pressure, stabilizing structures [2], and influencing the diffusion of large molecules [4].

Hyaluronan is one of the basic components of the extracellular matrix, in which it interacts primarily with a number of proteins [5]. The first interaction with proteins was discovered in 1972 [6]. Proteins capable of such interactions are called hyaladherins [7], e.g., HARE, which controls the endocytosis of hyaluronan [8,9]; LYVE1 [5,9], which is structurally similar to the CD44 receptor [9]; and many others including, intracellular receptors whose functions have not yet been fully clarified [10]. For most hyaluronan binding proteins, a domain of about 100 amino acids, known as the “Link module,” is responsible for binding [5,8,11].

CD44 is the major cell receptor for hyaluronan [12]. The part of CD44 called HABD is responsible for binding with hyaluronan [13]. Hyaluronan hexasaccharide (2.4 kDa) seems to be of the optimum size for binding to the CD44 receptor [14,15]. However, shorter fragments of between about 3 and 5 disaccharide units (1.2–2.0 kDa) also have receptor binding capacity [12]. The binding of hyaluronan to the receptor is mainly driven by their shape and their ability to form hydrogen bonds and van der Waals forces, rather than by electrostatic interactions [16,17,18]. Electrostatic interactions in physiological saline solution contribute to the total energy of the interaction by approximately 25%, although at physiological pH, almost all carboxyl groups are dissociated [6]; moreover, they are strongly dependent on the ionic strength of the environment [19]. A total of 13 amino acid residues are involved in the interaction between hyaluronan and CD44. A large part of the interaction is mediated by aliphatic, aromatic, and basic residues [16].

In addition to its binding to cellular receptors, where electrostatic interactions involving hyaluronan contribute only partially, the ability of hyaluronan to create electrostatic interactions has also become the basis for many other research studies.

The electrostatic binding of low-molecular-weight ligands to hyaluronan was studied by capillary electrophoresis [20]. It was found that β-naphthylamide derivatives with Arg, Lys, and Ala interact only very weakly in phosphate buffer (pH 7.4; I = 0.17 M), unlike in acetate buffer medium (pH 4.65; I = 0.05 M), where the electrostatic interaction was stronger mainly due to the lower ionic strength.

Oyarzun-Ampuero and his team [21] prepared nanoparticles by simply mixing solutions of hyaluronan (165 and 29 kDa) and polyarginine (5–15 kDa). It was found that systems can be prepared depending on the charge ratio between positive or negative zeta potential. At the same time, higher molecular weight hyaluronan was shown to form more stable particles. The stability of nanoparticles was tested as short term in PBS, as well as long term in water for 3 months. Systems with low-molecular-weight hyaluronan were shown to be unstable in PBS.

The electrostatic complexation of hyaluronan was also used for targeted siRNA distribution [22]. Hyaluronan (19 kDa) interacted with poly L-arginine (15–70 kDa) in aqueous media, and the resulting complex was used for interaction with siRNA and its targeted distribution. A simple mixing of the solutions was used to prepare the complexes.

A similar example of electrostatic interaction between hyaluronan and a cationic liposome/plasmid DNA complex was created by the Balbino group [23], where they allowed the liposome/DNA complex to interact with low-molecular-weight hyaluronan (16 kDa) in aqueous media in order to provide tools for targeted gene distribution.

Hyaluronan has already been shown to interact with amino acids under appropriate conditions [11]. In particular, pH and ionic strength appear to be important; when the pH drops to about 2.5, virtually no interactions can be observed, since the carboxyl groups of the hyaluronan providing the interaction are no longer dissociated [24]. According to calculations (density functional theory), Arginine together with hyaluronan form a more rigid structure, while lysine is more flexible [11].

The formation of complexes and aggregates between proteins and polysaccharides is often driven entropically in the case of strongly charged polyelectrolytes, probably through released counterions and water molecules and conformational changes in polymers during interaction [25].

Bovine serum albumin and hyaluronan have been shown to be capable of electrostatic complexation at around a pH 4, independently of the length of hyaluronan [26]. A single BSA molecule can be enveloped by multiple short HYA chains, while a long hyaluronan molecule can interact with multiple BSA molecules [26]. The turbidity of the solution and the size of the complexes were measured, and the HYA-protein complexes were analyzed by spectrophotometric measurement [26]. Complexes of hyaluronan and bovine serum albumin may be of three types [27]—namely, neutral insoluble complexes in the phase separation region and small positively charged complexes or large negatively charged complexes, which are soluble and are formed in an excess of albumin or hyaluronan, respectively.

Complexes of hyaluronan and silk fibrin (silk fibroin) have also been studied. These complexes exist in the pH range of 2.5–3.5, and the complexes are mainly formed by electrostatic interactions [27]. In contrast, lysozyme, which has an isoelectric point higher than albumin, formed complexes at significantly higher pH (pH 3–9) [28].

Thus, interactions between hyaluronan and amino acids, which also form proteins, are of interest from the point of view of both the functioning of hyaluronan in biological systems and its application in medical technologies and nanotechnologies. The aim of this work was to obtain a detailed picture of interactions between hyaluronan and various arginine oligomers using ultrasound spectroscopy and isothermal titration calorimetry, the latter also delivering basic thermodynamic parameters for these interactions. Such a system has not been investigated by these two techniques and only rarely by some other techniques [11].

## 2. Materials and Methods

### 2.1. Materials

Hyaluronan samples were obtained from Contipro Biotech (Dolní Dobrouč, Czech Republic), where they were produced by extraction from cell walls of the bacteria *Streptococcus zooepidemicus*. The product types are named according to the range of molecular weights in which the particular product falls. Seven product types were used in this work. Their molecular weights ranged from 9 to 1540 kDa. The various product types and the exact molecular weights of all used samples, as determined by the producer using size exclusion chromatography with multi-angle light scattering (SEC-MALS) technique, can be found in Appendix A.

Hyaluronan solutions were prepared at a concentration of 0.1% (weight, i.e., *w*/*w*, % are used throughout this work) by dissolving hyaluronan powder in ultrapure deionized water from a PURELAB purification system (OPTION R7/15; ELGA, High Wycombe, Great Britain), in PBS (VWR Life science, Radnor, PA, USA, Lot NO.: 0016C297, 137 mM sodium chloride, 2.7 mM potassium chloride, 12 mM phosphate buffer; ionic strength 172 mM), or in NaCl solutions of various concentrations (10, 50, 100, and 150 mM); both the hyaluronan powder and the solvents were weighed using analytical balances. The used hyaluronan concentration corresponded to a negative charge concentration of 2.5 mM at complete dissociation of the hyaluronan carboxyl groups. The mixture was stirred for 24 h in a closed vessel at room temperature to ensure complete dissolution [29].

Arginine hydrochlorides in monomeric and oligomeric forms were obtained from several suppliers, as shown in Appendix A. Arginine oligomers were synthesized commercially according to our requirements. Monomeric arginine hydrochloride was purchased from Sigma Aldrich (St. Louis, MO, USA) as a commercially available product.

Arginine solutions were prepared as 30 mM solutions of monomer units, by direct dissolution of a given arginine oligomer in volumetric flasks containing water, PBS, or NaCl solutions of various concentrations. The mixture was stirred for 24 h at room temperature to ensure complete dissolution. Preprepared NaCl or PBS solutions were used instead of water to prepare solutions of higher ionic strength.

Because the stock solutions of some of the arginine oligomers (dimer, tetramer, and octamer) had too low a pH (probably due to some excess of acid used to prepare the hydrochloride form) (see Appendix A), their values had to be adjusted so that their pH was above the hyaluronan dissociation constant [24] and similar to the pH of the other oligomers (around 6). The pH adjustment was done by adding a small volume of 2 M NaOH (G.R. Batch no. PP/2008/06964/0, Lach:ner, Neratovice, Czech Republic) to the stock solution (Appendix A). Although these pH adjustments led to an increase in the ionic strength, their effect was negligible. The increase in ionic strength during titrations with the pH-adjusted solutions was at most 8 mM, which had no observable effect on the titration curves, as demonstrated in independent experiments on the effect of the ionic strength. Arginine decamer stock solution had a pH of 5.71 ± 0.03, dodecamer, 5.62 ± 0.04, and triacontamer (30-mer), 6.77 ± 0.03.

### 2.2. Methods

#### 2.2.1. High-Resolution Ultrasound Spectroscopy

The ultrasonic velocity for each sample was measured at six selected frequencies in the range from 2.5 to 14.9 MHz using an HR-US 102T ultrasonic spectrometer (ultrasonic Scientific, Dublin, Ireland) with titration accessory. This device is equipped with two cells enabling single-cell or differential measurements. The differential regime was used in this work. It allows a resolution of 0.2 mm·s^−1^ for ultrasonic velocity (this corresponds to a value of 10^−5^% in water). The temperature was controlled with a Thermo Scientific Haake PC 200 heating bath, which provided a temperature stability of ±0.01 °C. The temperature was recorded by a sensor inserted in the ultrasonic spectrometer by the manufacturer. The measuring and reference cells were filled with deeply degassed sample solution and water, respectively, using a calibrated 1 mL Hamilton syringe. A volume of 1 mL was placed in each cell.

The measuring cell was filled with deeply degassed hyaluronan solution and solution containing amino acid was added through the titration accessory. The reference cell was filled with ultrapure and degassed deionized water. Ultrasonic velocity and attenuation were monitored continuously over the course of the titration at 25 °C.

Before each measurement, baseline correction was performed by taking measurements with deionized water in both cells. The measured dependences did not show any effect of the ultrasound frequency; consequently, only the results from measurements made at 11.6 MHz are reported.

Each measurement was made at least three times, and average values are reported in figures for the sake of clarity; error bars are shown in the figures. Reproducibility was much better in the case of relative velocity than in the case of attenuation in the sample cell, as expected.

From each amino acid titration to hyaluronan, the titrations of the solvent to hyaluronan and of the corresponding amino acid to the solvent were subtracted to account for dilution effects.

#### 2.2.2. ITC

Heat effects resulting from the titration of a titrant solution into the measuring cell were, in most cases, measured using a MicroCal PEAQ-ITC instrument (Malvern Panalytical Ltd., Westborough, MA, USA). A minority of experiments were performed with a Nano ITC 2G instrument (TA Instruments, New Castle, DE, USA). The titration of water to water was performed prior to each experiment to ensure that the instrument measured correctly. Each measurement was performed at least three times; average values are reported in figures and tables for the sake of clarity; standard deviations are given in tables and corresponding error bars are shown in figures. The reference cell was filled with water in both devices.

All experiments performed with the MicroCal PEAQ-ITC calorimeter had the reference power set to 41.9 µW. Peak integration and the processing of data obtained from the measurements were performed using MicroCal PEAQ-ITC Analysis Software (version 1.21, Malvern Panalytical Ltd., Westborough, MA, USA). In the case of Nano ITC 2G, data were processed using the NanoAnalyze Data Analysis program (version 2.1.13, TA instruments, New Castle, DE, USA). Both software use standard procedures [30] to determine basic thermodynamic parameters—specifically, here, the standard interaction Gibbs energy (Δ*G*), the standard interaction enthalpy (or the enthalpic contribution; Δ*H*), and entropy (Δ*S*); the product *T*Δ*S* is called the entropic contribution. The software also delivers an equilibrium constant (*K*), usually called the dissociation constant; this traditional terminology is kept here, although it is not exactly appropriate for the observed interactions.

When investigating interactions of arginine oligomers with hyaluronan, the hyaluronan solution (0.1 %wt) was, in all cases, placed in the cell and arginine oligomer solution was added. For MicroCal PEAQ-ITC, a 40 µL syringe was used. The first addition was 0.8 µL, other additions were 2 µL; a total of 36.8 µL of stock solution was added to 280 µL of hyaluronan stock solution. The stirring rate was set to 750 rpm and the interaction parameters were measured at 25 °C. The interval between additions was empirically set at 150 s, which was sufficient time for the signal to return to the baseline. Measurements were performed in pure water, PBS, or NaCl solutions of various concentrations (10, 50, 100, and 150 mM).

In the case of Nano ITC 2G, a 250 µL syringe was used with a first addition of 5.14 µL followed by 20 additional additions of 12 µL to a 990 µL cell. The stirring rate was set to 250 rpm, and the interaction parameters were measured at 25 °C. The interval between additions was empirically set at 500 s, which was sufficient time for the signal to return to the baseline.

To determine the dilution heats, titrations of arginine oligomer into water were performed. The heat of dilution was subtracted from the titration at the start of the evaluation. The dilution heat of hyaluronan during the experiment was not taken into account, because the heat released during one titration is in the order of hundreds of microjoule, while the dilution heat of hyaluronan is very small and, according to available literature [31,32,33], corresponded to the order of units of microjoule in water for our hyaluronan solutions.

#### 2.2.3. The Molar Ratio

The molar ratio was calculated as the ratio of the number of hyaluronan disaccharide units and the number of arginine monomer units present in solution. A molecular weight of 401.229 g/mol for hyaluronan disaccharide repeating units was used together with its weight-average molecular weight as determined by SEC-MALS. If the presence of one negative charge on each hyaluronan disaccharide unit and one positive charge on each monomer unit of arginine hydrochloride oligomers is assumed, the molar ratio gives the ratio of negative and positive charges. In other words, the molar ratio is then the ratio of the number of charges on hyaluronan to the number of charges on arginine.

## 3. Results

Basic information about the system was obtained by simple visual observation. Molecular weights of hyaluronan of 9 and 1540 kDa were tested with all arginine oligomers. All visual observations for solutions in water, PBS, and environments with increased ionic strength are summarized in Appendix A. No changes were observed for the monomer or for the dimer and tetramer of arginine hydrochloride during addition to the hyaluronan solution and the solutions remained clear. In the case of higher oligomers, the system generally changed from clear to opaque and subsequently cloudy for 9 kDa hyaluronan; in the case of 1540 kDa hyaluronan, there was also a macroscopic phase separation (precipitation). These phase changes generally occurred in water at a molar ratio around one. If additional ions were present in the solution, these changes generally occurred at higher molar ratios. Such shifts were particularly evident in the case of PBS.

### 3.1. Results from High-Resolution Ultrasound Spectroscopy

Arginine monomer and oligomers up to and including four monomer units showed almost unchanged ultrasound velocity during the whole titration course in water; data are shown in Appendix A for hyaluronan of the lowest and highest molecular weights used. This indicated no interactions, at least from the point of view of the ultrasound wave. As noted above, the pH of dimer and tetramer solutions had to be adjusted to a neutral value; the adjustment resulted in no change in the ultrasound spectroscopy record (cf. Appendix A).

The situation changed with arginine octamer in water. Visual observations showed that, unlike the shorter arginine oligomers, arginine octamer developed turbidity during titration. Observations varied depending on the used molecular weight of the hyaluronan. In the case of pH-adjusted solutions of arginine octamer, the change was clearly evident for the smallest molecular weight of hyaluronan. Unlike with the shorter arginine oligomers, the relative velocity decreased during titration as the hydration water was released from the hydration shell. This was particularly evident with high-molecular-weight hyaluronan, as shown in Figure 1. The decrease stopped at around a molar ratio equal to one, i.e., around the supposed charge equilibration, then the relative velocity slightly increased. The pH-adjustment, which was also made for the octamer, did not result in essential changes in titration profiles; see Appendix A. In comparison with pH-untreated samples, the decrease in the relative velocity was more distinct in the case of very low-molecular-weight hyaluronan, whereas it was not so deep for the high-molecular-weight sample. The pH-adjustment also “smoothed” the velocity decrease and, instead of a minimum, a sigmoidal shape with a point of inflection at a molar ratio of about 1.7–1.9 was observed.

The results for longer arginine oligomers (10, 12, and 30) were similar, and more detailed study of the effect of hyaluronan molecular weight was made using decamer and dodecamer. First, the relative velocity decreased almost linearly during the titration up to a molar ratio of about one. Afterwards, the velocity decreased more sharply and then, slightly increased (decamer and dodecamer) or remained almost constant (triacontamer). Data are shown in Figure 2 and in Appendix A. Here, also some effect of hyaluronan molecular weight can be seen. Hyaluronan with the highest molecular weight of 1540 kDa had the steepest slope in the initial titration phase while the lines of all the other samples were undistinguishable. Hyaluronan of the lowest molecular weight (9 kDa) gave the smallest drop in velocity during titrations. Further, the difference between the highest and lowest molecular weight of hyaluronan was smallest in the case of the arginine triacontamer (Appendix A).

The results of the visual observations for longer arginine oligomers are summarized in Appendix A. In the case of hyaluronan with the highest molecular weight tested, a cloudy solution was always formed, which turned into a precipitate at around a molar ratio of 1, after which the solution remained clear. At the lowest molecular weight of hyaluronan, a cloudy solution was formed with the decamer and dodecamer of arginine. As with the highest molecular weight of hyaluronan, the triacontamer of arginine formed a precipitate, but at this time, the solution remained slightly cloudy even after its formation.

#### Influence of the Environment

The longest arginine oligomers, i.e., dodecamer and triacontamer, were chosen for these experiments based on the results of measurements in water, in which they demonstrated interactions with hyaluronan of all molecular weights studied. The influence of ions or ion-controlled pH was studied by preparing solutions in PBS or in NaCl solutions of varying concentration.

The effect of PBS at pH 7.4 on interactions was investigated with arginine triacontamer. Selected molecular weights of hyaluronan in the range of 9–1540 kDa were tested, as shown in the graph in Figure 3. The relative velocity decrease did not occur from the first titration step as it did in water. The first additions resulted in constant values of relative velocities, so it is likely that interactions did not occur during this time. Then a drop in relative velocity occurred—as in the case of titrations in water—perhaps with the exception of 9 kDa hyaluronan, where the velocity decrease was smooth and moderate. The drop in the case of 109 kDa was not as steep as for the remaining samples of higher molecular weight and spread over a certain range of the molar ratio. In general, the molar ratio at which the drop occurred was shifted to higher values in comparison to the titrations in water. However, the effect of PBS was dependent on the hyaluronan molecular weight. This is also seen in Appendix A, where the increasing ultrasound attenuation well illustrates the (micro)phase separation and the formation of precipitates that occurred at higher molecular weights of hyaluronan. Increased attenuation means the formation of structures or aggregates capable of scattering the ultrasound wave and increasing sample heterogeneity [34,35]. Precipitates were observable for hyaluronan molecular weights from 310 kDa and were formed from a molar ratio of 1.5. Thus, particles were created in the system that allowed the sound to be dispersed. Greater dispersion (higher attenuation) occurred with higher molecular weight hyaluronan.

Arginine dodecamer in PBS were tested using ITC only; no heat effects were observed. Visual observations indicated that the solutions remained clear (Appendix A).

The effect of increasing NaCl concentration, i.e., of increasing ionic strength, was investigated using arginine dodecamer, which did not show interactions in PBS, together with hyaluronan of molecular weight 680 kDa. As shown in Figure 4, as the ionic strength increased, the drop in the relative velocity value gradually decreased. The slope of the initial decrease in the relative velocity also decreased with increasing ionic strength and, at the two highest NaCl concentrations, it was almost zero. Visual observations showed that in both cases, a precipitate occurred around a molar ratio of 1.1, but in the case of increased ionic strength (100 mM NaCl), the supernatant solution remained cloudy (Appendix A). At the highest NaCl concentration (150 mM), the drop in the relative ultrasound velocity was shifted to a significantly higher molar ratio but, at the same time, seems not to be statistically significant. This is in good agreement with the visual observation, which showed only very slight turbidity in 150 mM NaCl (Appendix A).

### 3.2. Results from ITC

ITC enabled a basic thermodynamic characterization of interactions between hyaluronan and arginine oligomers to be obtained. All measurements reported in this part were performed in an aqueous medium without buffer. After subtracting the dissolution heat from the heat obtained during titration, all performed titrations were found to be endothermic, as shown in Appendix A. Three types of titration curves (showing the dependence of the enthalpy change on the molar ratio during titration) were observed. Besides the sigmoidal shape typical for interacting systems with the saturation of interacting sites, simply decreasing and two-site curves were also obtained.

For the shortest arginine oligomers, including octamer, a typical sigmoidal titration pattern was not observed, irrespective of the hyaluronan molecular weight. As shown in Figure 5, the curves decreased steeply from the beginning of titrations up to the molar ratio of about one, from which no heat effects were recorded. These curves resembled the decreasing parts of typical sigmoidal ITC curves, which indicates the saturation of interaction sites and the end of interactions. Visually, dimer and tetramer systems remained clear, whereas octamer and low-molecular-weight hyaluronan systems were gently opaque at low molar ratios. The arginine octamer (pH untreated) and high-molecular-weight hyaluronan system also exhibited opacity, which did not disappear; a precipitate was even formed at around a molar ratio of one. These observations were not reflected in ITC curves, which were for all systems practically indistinguishable (cf. Figure 5). From these titration curves, only the overall change in the standard enthalpy change (Δ*H*; Appendix A) was calculated as the difference between the first and the last addition.

Typical sigmoidal titration shapes were obtained with arginine dodecamer and triacontamer and hyaluronan of all tested molecular weights. An example is shown in Figure 6. This shape points to hyaluronan–arginine interactions from the first oligomer additions in all investigated cases, which continued until the available hyaluronan interaction sites in solution were depleted at a molar ratio of about one. The sigmoidal shape was also observed for titrations of arginine octamer with adjusted pH into high-molecular-weight hyaluronan, with an inflex point shift to a molar ratio of about 1.5. This system is also represented in Appendix A to illustrate the baseline shift observed because of the change in the heat capacity of the system [36] in cases with the formation of a precipitate.

Using software implemented in the used apparatus, titrations with a typical sigmoidal shape were fitted with a traditional single binding site model to determine thermodynamic parameters (see Methods). The obtained parameters are shown in Appendix A. Thermodynamic energy-based binding (or, more precisely, interaction) parameters for various arginine oligomers are not appreciably different, and the ratios of the standard enthalpic and standard entropic contributions were very similar; an illustrative example is given in Figure 7 for the arginine dodecamer. Arginine oligomer–hyaluronan interactions were characterized by small and unfavorable standard enthalpy changes and were entropically driven. The smallest molecular weight of hyaluronan (9 kDa) had the highest dissociation constant; other molecular weights above 109 kDa had lower dissociation constants with approximately the same values. In addition, the arginine triacontamer had, overall, lower dissociation constant values compared to the arginine dodecamer. However, the dissociation constants were determined with relatively high standard deviations due to the high sensitivity of the model used for fitting to the steep changes in the titration curves from which the constants are determined. Such constants can thus be used only for the estimation of relative differences between different titrated systems.

From the viewpoint of isothermal titration calorimetry, the decamer was a unique oligomer. Its titration isotherm (see example in Figure 8) did not manifest either a sigmoidal or simple decreasing shape as in the case of the other investigated oligomers; its shape was, however, typical for the case involving several binding sites [30,37] (interaction events seem to be a more appropriate term in our case).

The fitting of multievent results was realized as follows. Data exemplified in Figure 8 were fitted by two models. The first model fitted the first, decreasing, part of the isotherm, the second model fitted the subsequent part with a maximum. The first part resembled the ITC curves measured with shorter arginine oligomers (cf. Figure 5) and was, therefore, called the “desolvation” model (desolvation part). The second model was called the “Two Sets of Sites” model in the ITC software and represents here some first and second interaction events. This multievent fitting varied with the molecular weight of hyaluronan. All three types of interactions were recognizable for the lowest molecular weight (Figure 8). However, as the molecular weight increased, the desolvation part disappeared and only the two remaining interaction events were retained (Appendix A). Note that the maximum on the two-interaction event model was still observed at the same molar ratio regardless of the hyaluronan molecular weight. The thermodynamic parameters estimated for each used hyaluronan molecular weight and model are shown in Appendix A.

#### Influence of the Environment

As in the case of ultrasonic spectrometer experiments, the longest arginine oligomers were chosen for experiments involving the presence of additional ions. Unlike arginine triacontamer, arginine dodecamer and decamer showed no interactions with hyaluronan in PBS (pH 7.4), regardless of the molecular weight of hyaluronan (no heat effects measured with ITC). In the case of arginine triacontamer, the titration curve was always observed to have a sigmoidal shape and the effect of PBS on interactions with hyaluronan of different molecular weights could be investigated with the single interaction site model. The estimated thermodynamic parameters are summarized in Appendix A. The molar ratio corresponding to the saturation of interacting sites (the inflex point on the ITC curve) increased with increasing hyaluronan molecular weight. The change in the standard interaction enthalpy was of one order of magnitude lower than in the case of the titration of solutions prepared only in water. As with aqueous solutions, interactions were dominated by the standard entropy contribution, as shown in Figure 9. The greater standard entropy contribution at hyaluronan molecular weights of 680 and 1540 kDa compared to that at other molecular weights of hyaluronan was due to the low value of the dissociation constant for these molecular weights, as shown in Appendix A.

Due to the inability of the arginine dodecamer to interact with hyaluronan in PBS, this arginine oligomer was chosen for testing the effect of increasing ionic strength. Several media with ionic strengths of 10, 50, 100, and 150 mM (NaCl concentration) were tested with 680 kDa hyaluronan. All samples except for the 150 mM ionic strength medium exhibited a sigmoidal titration curve. With 150 mM NaCl, however, the interactions were no longer noticeable (no heat effects measured with ITC). The data in Appendix A show that the molar ratio at the point of interaction saturation remained at about one and thus, no shift was observed as it was in PBS solution. The magnitude of the standard interaction enthalpy change decreased with increasing ionic strength, whereas the dissociation constant remained approximately constant (Appendix A). The change in standard Gibbs energy and the standard entropy contribution appeared to be unaffected by the magnitude of the ionic strength. Thus, the ionic strength did not appear to have a significant effect on the intensity of the interactions, unless it was sufficiently high (150 mM).

The arginine decamer, which demonstrated very specific behavior in ITC performed in water, did not show any interactions in ITC performed in PBS, irrespective of the molecular weight of hyaluronan. The solution also remained clear throughout the titration (Appendix A). Environments of 10 and 50 mM NaCl were also tested. In both cases, the decamer of arginine interacted similarly with hyaluronan of molecular weights 9 and 1540 kDa. It formed turbid systems or precipitates with low- or high-molecular-weight hyaluronan, respectively (Appendix A). All samples with high-molecular-weight hyaluronan exhibited a sigmoidal titration curve with yet smaller values for the standard interaction enthalpy than those measured for the arginine dodecamer in a similar environment (cf. Appendix A). In contrast, samples with low-molecular-weight hyaluronan exhibited only a simple decrease, similarly to shorter oligomers of arginine. From the titration curves with 9 kDa hyaluronan, only the overall change in the measured enthalpy change was calculated as the difference between the first and the last addition. Sigmoidal curves were evaluated by means of a standard model. The results of individual titrations are summarized in Appendix A. The specific behavior of the decamer was thus suppressed by the presence of additional ions.

## 4. Discussion

For arginine monomer and oligomers up to and including four monomer units, no interactions were observed in solutions prepared in water, even after pH adjustment, regardless of the used molecular weight of the hyaluronan. The relative ultrasound velocity did not change (Appendix A), the ITC curve simply decreased (Figure 5), and visual observation revealed a clear solution during the corresponding titrations, as seen in Appendix A presenting a basic overview of the results. A decreasing (and only decreasing) enthalpy change in ITC titrations is only very rarely reported in literature. A similar result was published [38] for the binding of two 9-O-(ω-amino) alkyl ether berberine analogues to the RNA triplex (though in an exothermic direction) but with no special comments—the authors simply considered it to be a part of the traditional sigmoidal curve. Taking into account the results of visual observation and ultrasound spectroscopy, we hypothesize that, in our case, the simple decreasing shape was probably the result primarily of some heat of mixing effects or of excess enthalpy (indicating nonideal behavior), which diminished with the increasing concentration of the hyaluronan–arginine mixture.

The situation began to change with arginine octamer. Although the ITC curves for the original (pH-untreated) samples still decreased simply for both studied molecular weights of hyaluronan (and were close to the curves obtained with shorter oligomers) (Figure 5), in the case of pH-adjusted solutions, a typical sigmoidal titration curve was obtained for high-molecular-weight hyaluronan (Appendix A). The relative ultrasound velocity was not constant in this latter case but decreased from the beginning of the titration, especially when the hyaluronan molecular weight was high (Appendix A). In the case of 1540 kDa hyaluronan, interactions marked by increased opacity and even some precipitate were also visually observable from a molar ratio of about one.

The typical sigmoidal ITC curve was obtained for arginine dodecamer and triacontamer (Appendix A) with hyaluronan of all tested molecular weights. The ITC records corresponded well with the data from ultrasound spectroscopy, as illustrated in Figure 10 for arginine dodecamer. The steep decrease in the sigmoidal curve corresponded to the sudden decrease in the relative ultrasound velocity. In contrast, ultrasound velocity was more sensitive to changes during the first part of titrations, where the ITC curve showed practically constant values of Δ*H* (the measured enthalpy change). Both techniques revealed some differences between the two different hyaluronan molecular weights.

A decrease in the relative ultrasound velocity usually indicates changes in hydration shells. Because the water in the hydration shell is less compressible than the bulk water [39], it is also denser [40], which helps sonic waves to propagate [34]; the relative velocity decreases when hydration water is released. Thus, the longer arginine oligomers disrupted the hyaluronan hydration shell to form an associate with a common hydration shell containing less hydration water molecules compared to the amount of hydration water in both shells separately. The relative velocity decreased until all corresponding interaction sites on the hyaluronan became saturated. Thereafter, these hydration changes no longer occurred, and the relative velocity started to increase only slightly. This increase can be attributed to the increasing amount of sodium and chloride counter-ions from hyaluronan and arginine, respectively, or to partial redissolution of the precipitate upon continuing dilution. It is interesting that these interactions were saturated at a molar ratio of about one, even though they are not strictly supposed to be electrostatic. The hydration changes might thus have occurred primarily on the charged groups and were induced by electrostriction effects [41,42,43,44].

The relative ultrasound velocity decrease in the first part of the titration was thus attributed mainly to hydration changes. More hydration water was released with high-molecular-weight hyaluronan as the relative velocity reached lower values.

A positive change in enthalpy was attributed previously also to the desolvation of some of the nonionic groups due to the interaction [45]. At the same time, new noncovalent bonds are not supposed to be formed during the interaction [46,47]. It is obvious from our ITC data that the entropic contribution to the Gibbs free energy was significant. Presumably, this large entropy gain arises also from the release of water molecules during titration [48]. Thus, our data indicate an important effect of the destruction of the hydration shell upon hyaluronan–arginine interactions. Moreover, the relationship between the (steep) changes in the titration curves and the molar ratio points to the involvement of charged groups and electrostatic forces (and electrostriction) in these interactions.

Further, the combined influence of arginine and hyaluronan molecular weights was observed with arginine dodecamer and triacontamer. Lower values of the relative ultrasound velocity were observed for 1540 kDa hyaluronan with the dodecamer in comparison to 9 kDa hyaluronan with the dodecamer. This means a higher amount of water released from the high-molecular-weight hyaluronan. This agrees with the lower enthalpy change and lower entropy contribution found for the 9 kDa hyaluronan (Appendix A). The situation is similar for arginine triacontamer but, here, 9 kDa hyaluronan led to a more gentle decrease in the ultrasound titration curve (Appendix A). Nevertheless, for both forms of hyaluronan, the intensive development of turbidity was observed during titrations with arginine triacontamer, finally resulting in the formation of a precipitate around a molar ratio equal to one.

The greatest difference between the two used techniques appeared to arise in the case of arginine decamer. Although ultrasonic spectroscopy did not distinguish between decamer and higher arginine oligomers, in the case of ITC, decamer revealed specific results that did not share the features of other arginine oligomers. The unique results obtained in ITC for arginine decamer still corresponded in their principal features to ultrasound titrations, and the effect of hyaluronan molecular weight was similar to that found for dodecamer. Figure 11 shows that the maximum on the ITC curve, attributed to the transition between the first and second interaction events, corresponded to the steep decrease in the relative ultrasound velocity record. The first, decreasing, part of the ITC curve thus corresponded to the ITC curves for shorter arginine oligomers in Figure 5 and to the first decreasing branch of the ultrasound titration curve. The latter was interpreted in terms of decreasing hyaluronan hydration. Even in this titration region, interactions could be observed visually (Appendix A) as a certain degree of opacity, the intensity of which increased with hyaluronan molecular weight. In the case of 1540 kDa, a precipitate even formed around a molar ratio of 1.2, which is around the maximum of the ITC curve in Figure 11. Thus, the similarity between the decreasing parts at low molar ratios in Figure 5 and Figure 11 is only apparent—they reflect different processes (with different Δ*H* values).

The two methods employed in this work do not provide sufficient chemical information to explain the specific thermal behavior of arginine decamer interacting with hyaluronan. Cationic side chains of arginine oligomers were found to have the ability to form like-charged contact ion pairs in an aqueous environment [49]. Although the ability to form like-charged contact ion pairs in the case of arginine is not limited by the number of monomers in the chain, it could be conducive with respect to establishing the different behavior of the arginine decamer—we speculate that just 10 arginine monomers result in a specific dimension and shape on the part of the corresponding oligomer, which is favorable for its specific interactions with hyaluronan. This length requirement mirrors a similar requirement in the binding of hyaluronan to the primary surface cell receptor CD44, which is found to require a hyaluronan sequence of at least 3–9 disaccharide units (1.2–3.6 kDa) for monovalent binding to occur [12,50]. Lysine does not have the ability to form like-charged contact ion pairs. We, therefore, conducted a pilot experiment with lysine decamer, in which no multievent ITC record was found (data not shown). Thus, the specific conformations of different arginine oligomers and hyaluronan and the structures of their hydration shells should be revealed to explain the observed interaction behavior. A molecular dynamic modelling study has recently been initiated to address this problem. 

Of all the studied hyaluronan molecular weights in water, the standard entropic contribution (and also the standard interaction Gibbs energy) of the lowest studied molecular weights of hyaluronan (9 or 16 kDa), which were in the rod conformation [51], had the lowest absolute values for both dodecamer and triacontamer arginine (Appendix A). The enthalpic contribution increased slightly with increasing hyaluronan chain length. For the distinctive arginine decamer, the desolvation model was only observable for 9 kDa hyaluronan and disappeared with increasing molecular weight, leaving only the model of the Two Sets of Sites. From the point of view of ultrasonic spectroscopy, 9 kDa hyaluronan achieved the smallest change in relative ultrasound velocity of all the studied molecular weights of hyaluronan for all arginine oligomers. Thus, it is likely that the rod conformation of hyaluronan releases less water during interactions [48]. In contrast, the other studied molecular weights of hyaluronan in random coil conformation [52] have approximately the same entropy contribution. Hyaluronan with a molecular weight of 109 kDa, which has a conformation close to the transition between the rod and random coil forms [53], appears, from the microcalorimetric point of view, more like a random coil in terms of its interactions with arginine oligomers.

Arginine interactions with hyaluronan with the shortest chains also showed the highest sensitivity to the presence of added ions—they were most suppressed by increased ionic strength.

As far as the effect of the molecular weight of hyaluronan is concerned, the occurrence of the rod or random coil conformation appeared to be crucial. The forms of hyaluronan in the region of the rod conformation (9 and 16 kDa) differed significantly from longer chains of hyaluronan with respect to individual parameters of ITC and HR-US. The conformation of the hyaluronan chain is thus relevant for interactions with arginine and its oligomers as well as their sensitivity to ionic strength.

From the point of view of the effect of ions, there was very good correlation between the results of both techniques both in PBS solution and environments with different ionic strengths. Figure 12 illustrates the combined effect of PBS and hyaluronan chain length on titration curves obtained by the two methods. The saturation point was shifted to a molar ratio of about 1.5 for high-molecular-weight hyaluronan, and it is also clearly evident that the interactions were much less intense for 9 kDa hyaluronan, consistent with respect to both ITC and HR-US results. In PBS and for arginine triacontamer, ITC also detected increasing absolute values for the standard interaction Gibbs energy and entropic contribution with increasing hyaluronan molecular weight up to 680 kDa (Figure 9); the enthalpic contribution increased slightly.

In environments with increasing ionic strength, there were no significant changes in the titration curves, except for 150 mM NaCl (Figure 4 and Appendix A). In 150 mM NaCl, the HR-US titration curve showed a very small and statistically insignificant change around a molar ratio of 1.7 (Figure 4), while no heat effects were detected by ITC (Appendix A). However, with increasing ionic strength, there was a decrease in the change in enthalpy (Appendix A) as well as a decrease in the absolute value of the minimum relative ultrasound velocity, as shown in the graphs in Appendix A. A common feature of both PBS and environments of varying ionic strength is a significant decrease in the size of the reaction enthalpy in comparison with aqueous solutions. Interactions in water are most intense, releasing the largest amount of hydration water and thus having the lowest relative ultrasound velocity value. With increasing ionic strength, the amount of hydration water released gradually decreases.

Generally, a sufficiently high ionic strength suppresses interactions between hyaluronan and arginine oligomers which occur in water. As the chain length of the arginine oligomer increases, the ability to better resist the influence of ions in the environment increases (the triacontamer is able to interact in PBS, whereas the dodecamer is only able to interact in a NaCl solution with a maximum concentration of 100 mM). The molecular weight of hyaluronan contributes in a similar way, and as its value increases, the interactions in the given environment are strengthened, as can be seen from the results of titrations of the decamer in environments with different ionic strengths (Appendix A).

The results of this study can be summarized as follows. In water, a sufficient length of arginine oligomer is necessary for it to be able to interact with hyaluronan, regardless of the molecular weight of hyaluronan. The transition from noninteracting to interacting oligomers occurs between octamer and dodecamer, with unique behavior observed for decamer. This is observed in the ultrasound titration curve by a change from an almost straight line (ultrasound velocity independent of molar ratio), by a slightly bent sigmoid, up to an approximately Z-shaped curve typical for a strongly interacting system. In ITC, a simply and steeply decreasing curve changes via an approximate Λ shape with a strong maximum (just arginine decamer) to the classical sigmoidal record. Visually, this transition corresponds to the change from a clear, through a turbid, to a precipitated system.

Hydration forces (relating to the release of water molecules from the hyaluronan hydration shell) play an important role in interactions. The microphase (turbidity) or even macrophase (precipitation) observed in some systems could have analogical desolvation (dehydration) causes such as the well-known salting out of proteins. Moreover, the electrostatic component is not negligible, as demonstrated by the saturation of interaction sites around the point of charge equilibration, which is in contrast to hyaluronan-CD44 interactions mediated through amino acid sequences on this receptor. The significance of these electrostatic forces was demonstrated also by the effect of ions from background electrolyte—the interactions were suppressed by a sufficiently high ionic strength (at least at a value of around 150 mM, i.e., the ionic strength of physiological solution). Very probably, hydration forces were combined with electrostatic forces forming electrostriction effects.

However, the suppression of interactions by the added ions could be eliminated by a sufficient length of arginine oligomer (for instance, 30-mer). Thus, hydrogen bonding and hydrophobic interactions between nonpolar parts of the interacting (macro)molecules also play a role, especially when their hydration shells are removed or disturbed. For hyaluronan, the existence of hydrophobic patches is described [54]; the hydrocarbon parts of arginine oligomer chains are hypothesized to form similar sites for hydrophobic interactions. Like-charge arginine pairing and the specific arginine properties behind it are probably responsible for the unique behavior observed for arginine decamer. 

Finally, whether hyaluronan occurs in rod-like or coiled conformation is also relevant for its interactions with arginine as well as for their sensitivity to ionic strength.

These interpretations are consistent with published concepts on the receptor binding of hyaluronan, which occurs in ionic physiological environment. The binding of hyaluronan to the CD44 receptor is controlled primarily by the shape of both—though mainly of the CD44 receptor—and their ability to form hydrogen bonds. A total of 13 amino acid residues are involved in the interaction with hyaluronan [16]. Hydrophobic interactions are of secondary importance [17]. Ionic interactions in saline contribute to approximately 25% of the total interaction energy and are highly dependent on the magnitude of the ionic strength [19].

## 5. Conclusions

The aim of this work was to investigate the interactions between arginine oligomers in hydrochloride form and hyaluronan. Interactions were investigated using isothermal titration calorimetry and high-resolution ultrasound spectroscopy. In water, no interactions were observed for the arginine dimer and tetramer. The transition from noninteracting to interacting oligomers occurred between octamer and dodecamer, with unique behavior observed for decamer. Dodecamer and triacontamer demonstrated clear interactions using both techniques. In water, a sufficient length of arginine oligomer is thus necessary for it to be able to interact with hyaluronan, regardless of the molecular weight of hyaluronan. Both techniques showed the saturation of interaction sites at molar ratios of basic units (arginine monomer and hyaluronan dimer) of around one (at the point of supposed charge equilibration). The interactions were suppressed by a sufficiently high ionic strength (a value of at least 150 mM or thereabouts, i.e., the ionic strength of physiological solution), providing the arginine oligomer was not too long (e.g., 30-mer) or the hyaluronan chain not too short (e.g., 9 kDa). Whether hyaluronan conformations were rod-like or coiled was important for hyaluronan’s interactions with arginine oligomers as well as for their sensitivity to ionic strength.

The two techniques gave generally corresponding outcomes, while in some cases, these differed in their details. Interactions between arginine oligomers and hyaluronan were explained by means of the combination of effects (forces). That is, hydration forces (the release of water molecules from the hyaluronan hydration shell), electrostatic interactions combined with electrostriction, as well as hydrogen bonding or hydrophobic contacts between the desolvated parts of (macro)molecules all play a role together with specific conformations of hyaluronan and arginine oligomers (which subsume the pairing ability of arginine). The employed techniques did not provide direct chemical information—molecular modelling could provide additional explanations and support for our conclusions.

## Figures and Tables

**Figure 1 polymers-12-02069-f001:**
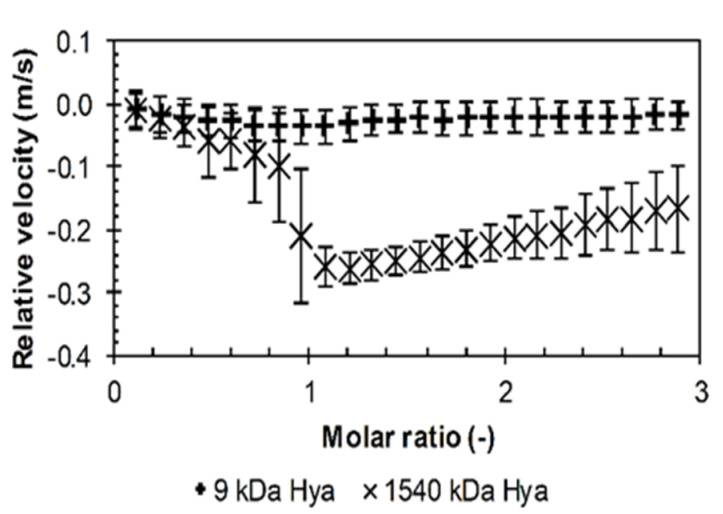
The relative ultrasonic velocity in dependence on molar ratio for titrations of arginine octamer in hydrochloride form to hyaluronan of different molecular weights in water (11.6 MHz, 25 °C).

**Figure 2 polymers-12-02069-f002:**
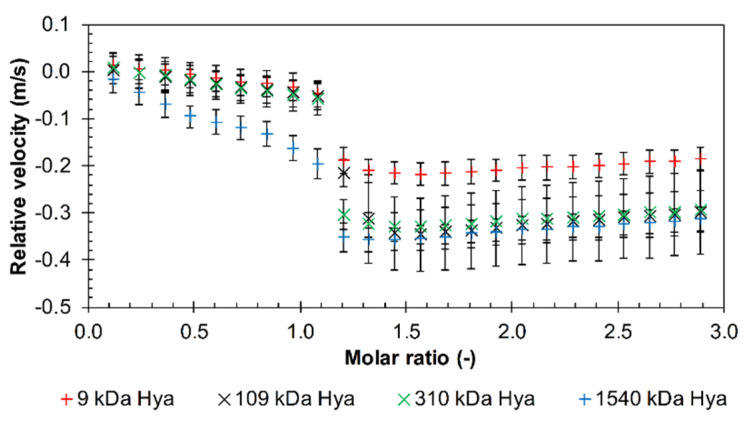
The relative ultrasonic velocity in dependence on molar ratio for titrations of arginine decamer in hydrochloride form into hyaluronan of different molecular weights in water (11.6 MHz, 25 °C).

**Figure 3 polymers-12-02069-f003:**
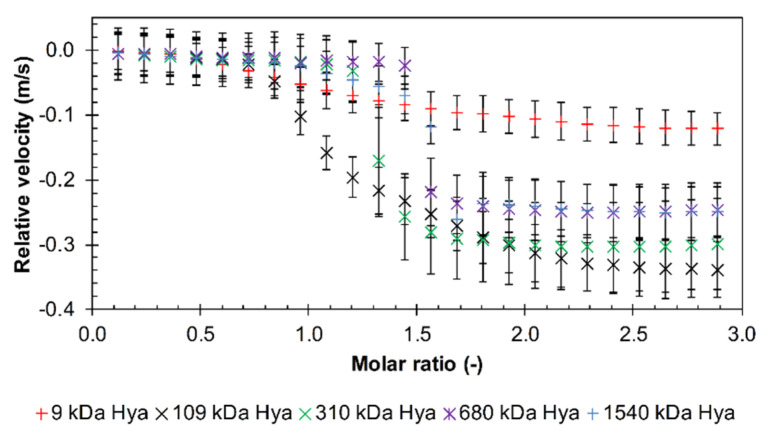
The relative ultrasonic velocity in dependence on the molar ratio for titrations of arginine triacontamer in hydrochloride form into hyaluronan of different molecular weights in PBS (11.6 MHz, 25 °C).

**Figure 4 polymers-12-02069-f004:**
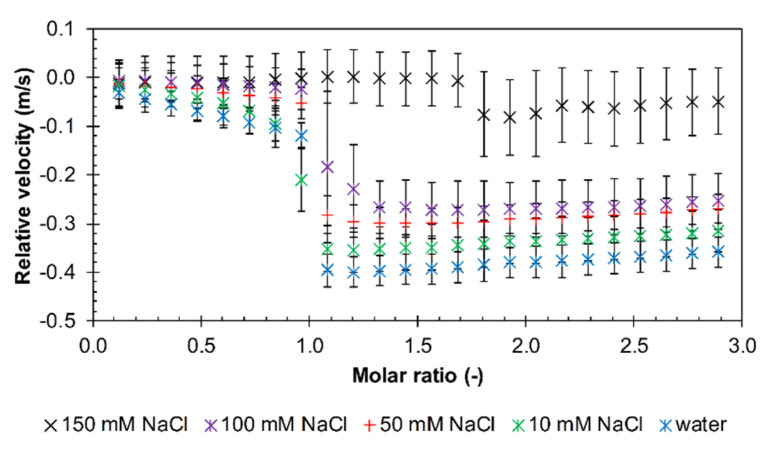
The relative ultrasonic velocity in dependence on the molar ratio for titrations of arginine dodecamer in hydrochloride form into 680 kDa hyaluronan solutions in different ionic strength environments (11.6 MHz, 25 °C).

**Figure 5 polymers-12-02069-f005:**
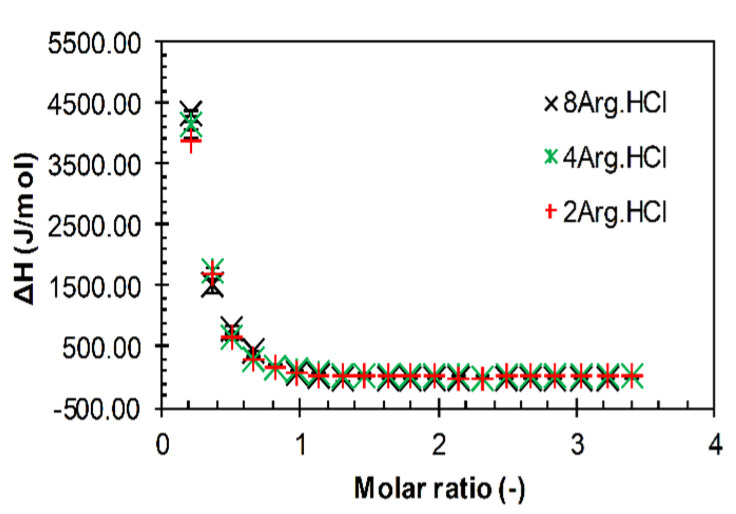
Isothermal titration calorimetry (ITC) records for the titration of various arginine oligomers in hydrochloride form into a 1540 kDa hyaluronan solution in water (25 °C).

**Figure 6 polymers-12-02069-f006:**
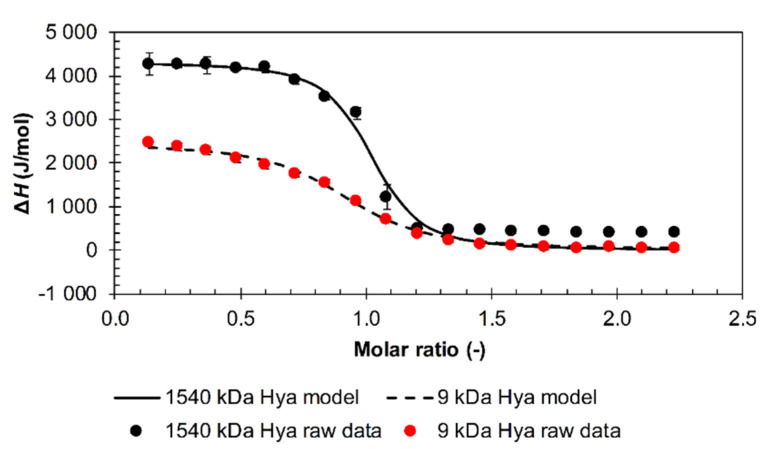
ITC records for the titration of arginine dodecamer in hydrochloride form into 9 and 1540 kDa hyaluronan solutions in water (25 °C). Single binding site models were used for data evaluation.

**Figure 7 polymers-12-02069-f007:**
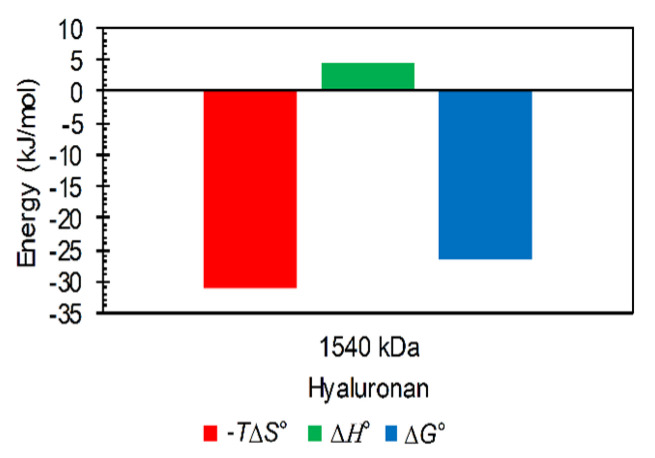
Example of magnitudes of thermodynamic parameters determined from ITC for interactions of arginine dodecamer in hydrochloride form with hyaluronan of molecular weight 1540 kDa in water (25 °C).

**Figure 8 polymers-12-02069-f008:**
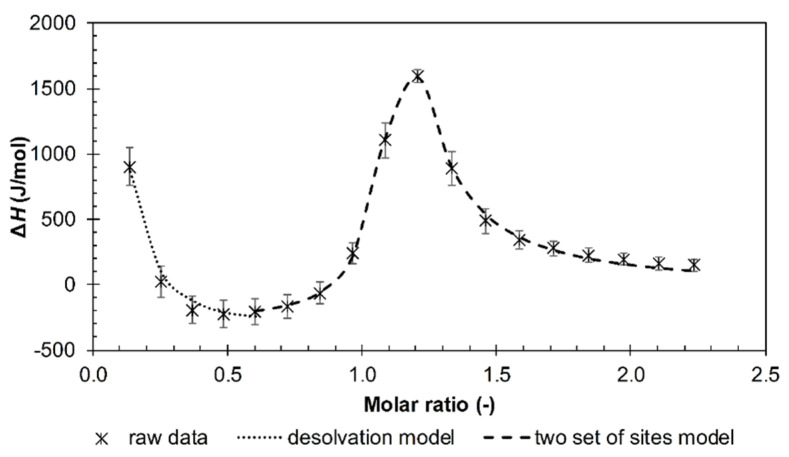
ITC records for the titration of arginine decamer in hydrochloride form into a 9 kDa hyaluronan solution in water (25 °C). Two sets of sites and the desolvation model were used for data evaluation.

**Figure 9 polymers-12-02069-f009:**
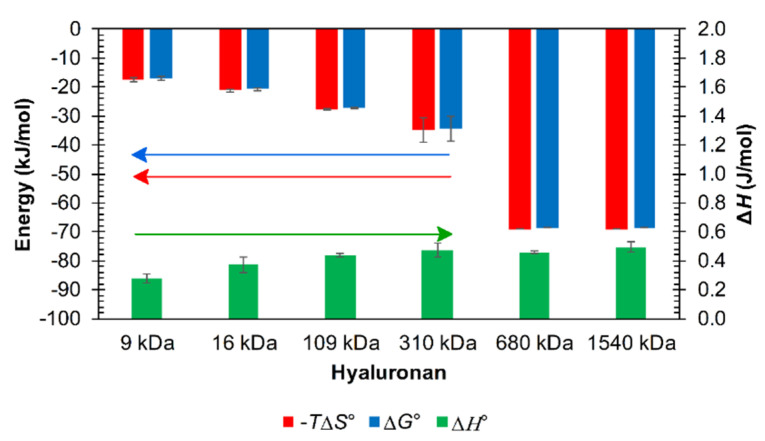
Graph showing the thermodynamic parameters obtained from ITC for the arginine triacontamer in hydrochloride form and hyaluronan of various molecular weights in the environment of PBS.

**Figure 10 polymers-12-02069-f010:**
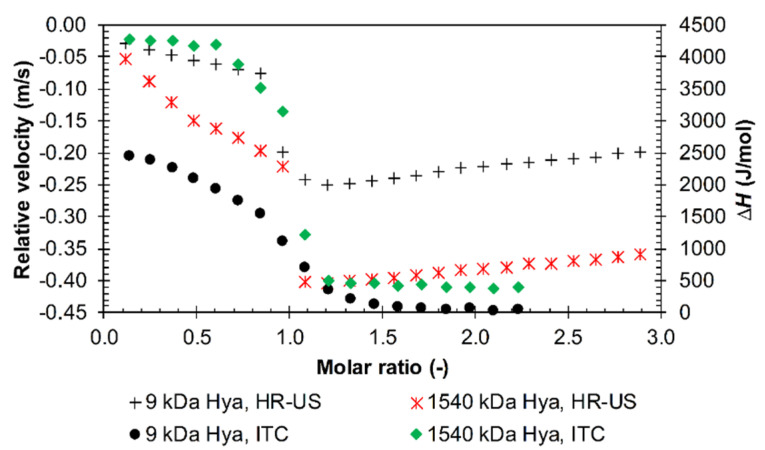
The comparison of HR-US (high-resolution ultrasound spectroscopy) and ITC titration records during the titration of arginine dodecamer in hydrochloride form into 9 and 1540 kDa hyaluronan solutions in water (11.6 MHz, 25 °C).

**Figure 11 polymers-12-02069-f011:**
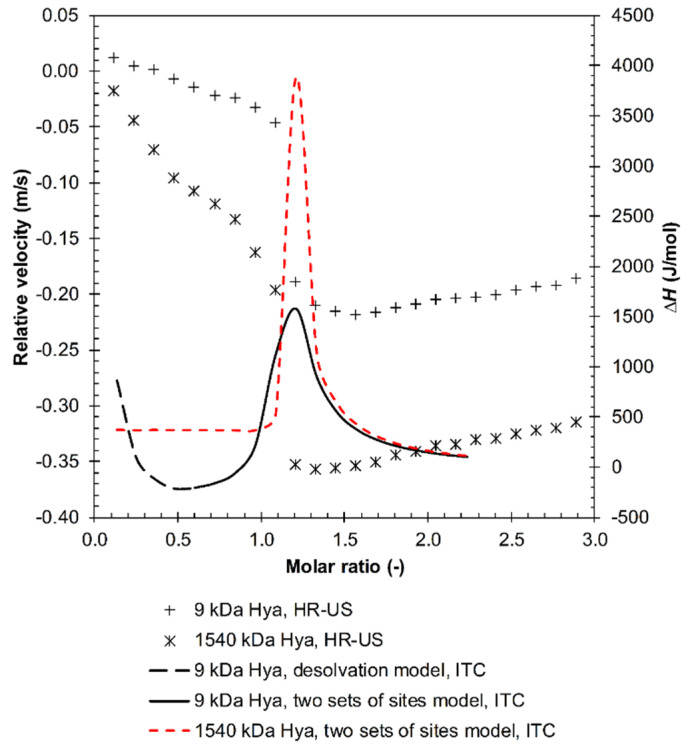
Comparison of the fitted ITC records and the measured relative ultrasound velocity changes during the titration of arginine decamer (hydrochloride form) into 9 and 1540 kDa hyaluronan solutions in water (25 °C, 11.6 MHz).

**Figure 12 polymers-12-02069-f012:**
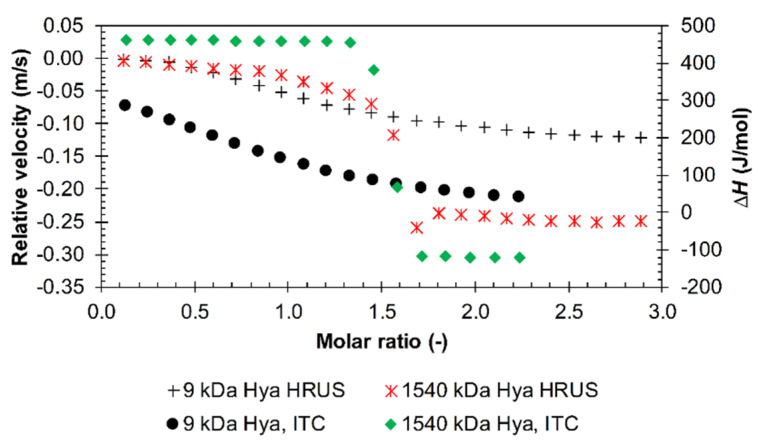
The comparison of HR-US and ITC titration records during the titration of arginine triacontamer (hydrochloride form) into 9 and 1540 kDa hyaluronan solutions in PBS (25 °C, 11.6 MHz).

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
