# Peer review of "Hyaluronan-Arginine Interactions—An Ultrasound and ITC Study"

_polymers, 2020, doi:10.3390/polym12092069_

Round 1
Reviewer 1 Report
Dear Editor,
This contribution by Adam Jugl and Miloslav Pekař reports a detailed investigation of the interaction between arginine oligomers and hyaluronan of different molecular weights, studied by high-resolution ultrasound spectroscopy and isothermal titration calorimetry. Hyaluronan is highly recognized as an important biopolymer found in extracellular matrix and therefore, advancing in the fundamental understanding of the interactions with other biological components is of relevance.
The manuscript is well organized, showing an extensive introductory section, as well as results clearly presented and properly discussed. Of course, as the authors acknowledged, the two methods employed in this work do not provide direct chemical information to explain the specific interactions leading to the thermal behavior observed. Nevertheless, authors were able to demonstrate how the length of arginine oligomers and some environmental factors (pH and salts) affected interactions with hyaluronan, including an extensive thermodynamic characterization. Given the relevance of hyaluronan in biological systems, this article should attract interest from the general reader in this journal. I definitely recommend this manuscript to be considered for publication in Polymers.
Author Response
Thank you for your comment.
Reviewer 2 Report
This paper describes various biophysical studies on hyaluronan, a topic outside my own field. It is an interesting and well-written manuscript, suitable for publication. The wording could be condensed a little in places (eg lines 61-64) but is very clear throughout.
The authors use either MW or degree of oligomerization to describe hyaluronan, and it would be helpful if these were both given in the Introduction itself and not just the SI. For example, HABD of CD44 binds a hexamer of hyaluronan - but how does this compare with the 9kDa hyaluronan later used in the experiments? Also this system could be used to explain briefly the preferred interactions hyaluronan makes with proteins.
Results (line 203). The MW of the hyaluronan is given as an average, but it is also important to know the distribution. Some indication of the spread of MWs in each sample if known.
I suggest indicating in brackets after the first use of the word triacontamer that it means 30mer.
I suggest that the buffer (if any) used for the ITC experiments be stated clearly at the start of Section 3.2.
A linear decrease in enthalpy with injection in ITC is indicative of non-ideality, and is often observed if little published. I am not sure what "excess enthalpy" is. From memory, lysine gives a much larger effect than arginine on dilution.
Precipitation can lead to strong heat effects in itself, which makes analysis of ITC data difficult. The unique heat signature of the arginine decamer is interesting, but probably reflects changes in the oligomers themselves rather than two distinct binding events. The authors note that their data are insufficient to allow very much to be deduced in terms of molecular models. This cautious approach is to be commended, and hopefully molecular modelling will give more clues as the the interaction.
The structural transition of the hyaluronan, mentioned in the Abstract, could be elaborated slightly in the Conclusions, I am not familiar with this system myself. Does ionic strength control this feature of the molecule?
Lysine and arginine have very different side-chains, arginine often making apolar interactions, and showing a surprising solubility in octanol. Overall it seems that the interactions at low ionic strength are non-specific (gel filtration of proteins is usually carried out with at least 100 mM NaCl to prevent such effects).
Author Response
- The authors use either MW or degree of oligomerization to describe hyaluronan, and it would be helpful if these were both given in the Introduction itself and not just the SI. For example, HABD of CD44 binds a hexamer of hyaluronan - but how does this compare with the 9kDa hyaluronan later used in the experiments? Also this system could be used to explain briefly the preferred interactions hyaluronan makes with proteins.
Information on the molecular weights of hyaluronan and their relationship to its polymerization degree was added (lines 37, 39 and 529). The polymerization degree can also be calculated using the basic unit molecular weight reported in manuscript. The interactions and their nature were discussed in conclusions and at the end of discussion, these interactions can be expected for arginine-containing proteins.
- Results (line 203). The MW of the hyaluronan is given as an average, but it is also important to know the distribution. Some indication of the spread of MWs in each sample if known.
Information on polydispersity was added to the Supplementary Materials, below Table S1.
- I suggest indicating in brackets after the first use of the word triacontamer that it means 30mer.
Done; line 133.
- I suggest that the buffer (if any) used for the ITC experiments be stated clearly at the start of Section 3.2.
Added sentence on line 308, specifying the use of buffer. In chapter 3.2.1. Influence of the environment is the used buffer specified on line 387.
- A linear decrease in enthalpy with injection in ITC is indicative of non-ideality, and is often observed if little published. I am not sure what "excess enthalpy" is. From memory, lysine gives a much larger effect than arginine on dilution.
We agree and, in fact, have the same thoughts in our minds. The "excess enthalpy", as an example of excess quantity known in physical chemical, is just a physico-chemical indicator of non-ideality of a mixture. Explained on line 443. - Precipitation can lead to strong heat effects in itself, which makes analysis of ITC data difficult. The unique heat signature of the arginine decamer is interesting, but probably reflects changes in the oligomers themselves rather than two distinct binding events. The authors note that their data are insufficient to allow very much to be deduced in terms of molecular models. This cautious approach is to be commended, and hopefully molecular modelling will give more clues as the the interaction.
We agree. We hypothesized in our manuscript that the changes in the case of arginin decamer are caused by the distraction of contact ion pairs on the side chains of arginine oligomers. We believe that this is for the first time this behavior was observed. As our techniques did not provide chemical analysis information we initiated the computer modeling study to reveal composition-specific causes. - The structural transition of the hyaluronan, mentioned in the Abstract, could be elaborated slightly in the Conclusions, I am not familiar with this system myself. Does ionic strength control this feature of the molecule?
The structural transitions are mentioned in the abstract and also in the discussion (lines 536-550). More details are given in referenced publication.
- Lysine and arginine have very different side-chains, arginine often making apolar interactions, and showing a surprising solubility in octanol. Overall it seems that the interactions at low ionic strength are non-specific (gel filtration of proteins is usually carried out with at least 100 mM NaCl to prevent such effects).
Thank you for your comment, it is consistent with our results.